# Qualitative Differences in Protection of PTP1B Activity by the Reductive Trx1 or TRP14 Enzyme Systems upon Oxidative Challenges with Polysulfides or H_2_O_2_ Together with Bicarbonate

**DOI:** 10.3390/antiox10010111

**Published:** 2021-01-14

**Authors:** Markus Dagnell, Qing Cheng, Elias S.J. Arnér

**Affiliations:** 1Division of Biochemistry, Department of Medical Biochemistry and Biophysics, Karolinska Institutet, 171 77 Stockholm, Sweden; qing.cheng@ki.se; 2Department of Selenoprotein Research, National Institute of Oncology, 1122 Budapest, Hungary

**Keywords:** PTP1B, redox regulation, bicarbonate, peroxymonocarbonate, polysulfide, TrxR1, TRP14 and Trx1

## Abstract

Protein tyrosine phosphatases (PTPs) can be regulated by several redox-dependent mechanisms and control growth factor-activated receptor tyrosine kinase phosphorylation cascades. Reversible oxidation of PTPs is counteracted by reductive enzymes, including thioredoxin (Trx) and Trx-related protein of 14 kDa (TRP14), keeping PTPs in their reduced active states. Different modes of oxidative inactivation of PTPs concomitant with assessment of activating reduction have been little studied in direct comparative analyses. Determining PTP1B activities, we here compared the potency of inactivation by bicarbonate-assisted oxidation using H_2_O_2_ with that of polysulfide-mediated inactivation. Inactivation of pure PTP1B was about three times more efficient with polysulfides as compared to the combination of bicarbonate and H_2_O_2_. Bicarbonate alone had no effect on PTP1B, neither with nor without a combination with polysulfides, thus strengthening the notion that bicarbonate-assisted H_2_O_2_-mediated inactivation of PTP1B involves formation of peroxymonocarbonate. Furthermore, PTP1B was potently protected from polysulfide-mediated inactivation by either TRP14 or Trx1, in contrast to the inactivation by bicarbonate and H_2_O_2_. Comparing reductive activation of polysulfide-inactivated PTP1B with that of bicarbonate- and H_2_O_2_-treated enzyme, we found Trx1 to be more potent in reactivation than TRP14. Altogether we conclude that inactivation of PTP1B by polysulfides displays striking qualitative differences compared to that by H_2_O_2_ together with bicarbonate, also with regard to maintenance of PTP1B activity by either Trx1 or TRP14.

## 1. Introduction

Induction of phosphorylation cascades during growth factor-dependent receptor activation controls cellular phenotypes such as differentiation, proliferation, and migration [1]. Protein tyrosine kinases (PTKs) and protein tyrosine phosphatases (PTPs) are key enzymes responsible for controlling receptor-linked phosphorylation cascades, and their dysregulation can lead to pathologies such as cancer or inflammatory diseases. The activities of PTPs rely upon an active site Cys residue that can be reversibly inhibited through oxidation. Because the oxidized Cys residue can be reactivated by reduction, this transient mode of regulation is a mechanism that enables signaling cascades to take place and this active site Cys residue of classical PTPs is well conserved, with its low pKa rendering it especially susceptible to oxidation [1,2,3,4]. Growth factor receptor-dependent induction of transient bursts of H_2_O_2_ by activation of NADPH oxidases (NOXs) [5,6] leads to reversible oxidation and thereby inhibition of PTPs, which allows for protein phosphorylation cascades to occur [4,7]. The membrane-bound NOXs produce extracellular superoxide, which is rapidly converted to H_2_O_2_ and brought into the cytosol through aquaporins [8,9]. In addition, mitochondrially produced H_2_O_2_ via p66Shc has been shown to contribute to PTP inactivation during platelet derived growth factor (PDGF) receptor signaling [10]. Recently we showed that for PTP1B oxidation to take place during epidermal growth factor (EGF) stimulation, the presence of cellular bicarbonate (HCO_3_^−^) is a necessity [11]. Previous work had identified that PTP1B oxidation by H_2_O_2_ could be potentiated by HCO_3_^−^ and was proposed to involve formation of the more reactive peroxymonocarbonate species [12]. Indeed, a direct reaction between H_2_O_2_ and HCO_3_^−^ leads to a low rate of spontaneous formation of peroxymonocarbonate [13,14] and PTP1B itself may possibly further facilitate this process [12].

Several different forms of oxidative modifications of the active site Cys of PTP1B have been described, including a sulfenic acid (-SOH) [3,7,10,15] that can rapidly undergo formation of a sulfenylamide bond with the peptide backbone [16,17,18] or lead to glutathionylation [19,20]. Recently, per- and polysulfide species have also been shown to modulate PTP1B activity, proposed to protect the enzyme from further irreversible oxidation that may otherwise occur by formation of sulfinic or sulfonic acid derivatives of the Cys residue, and exogenously added polysulfides prior to stimulation of the EGF receptor (EGFR) was found to potentiate EGFR phosphorylation [21,22]. The potency and effects of polysulfides on PTP1B compared to the effects of H_2_O_2_ and HCO_3_^−^ however need further characterization, and should also be considered in view of the reduction and reactivation of oxidized PTP1B by the reductive enzyme systems. It is well known that different oxidized forms of the PTP1B can be reduced by different components of the thioredoxin (Trx) system. For example, the sulfenic acid form is potently reduced directly by Trx reductase 1 (TrxR1), but this enzyme cannot directly reduce the sulfenylamide form of PTP1B; that reduction requires either Trx1 or Trx-related protein of 14 kDa (TRP14) [11,23,24,25,26]. Interestingly, knockdown of TRP14 in HEK293 cells further increases EGF-dependent EGFR phosphorylation that is accentuated by exogenous polysulfide addition, suggesting a regulatory function of per- or polysulfidation of PTP1B counteracted by TRP14 during EGF signaling [22]. We recently studied the effects of H_2_O_2_ and HCO_3_^−^ on PTP1B activity in the presence of a complete Trx system coupled with peroxiredoxin 2 (Prx2) using pure recombinant proteins, showing that addition of HCO_3_^−^ could potently overcome the complete reductive protection of PTP1B by the Trx system against the addition of H_2_O_2_ alone [11,26]. In the present study, we aimed to characterize the effects of polysulfides on PTP1B activity and compare those with the inhibition of the enzyme by H_2_O_2_ and HCO_3_^−^, also comparing the protective effects of TRP14 in this context in relation to those exerted by Trx1. We found that both Trx1 and TRP14 potently protect PTP1B from inactivation by polysulfides, but less so from H_2_O_2_ in combination with HCO_3_^−^, and we found that Trx1 is more potent than TRP14 in reactivating oxidized PTP1B, especially upon prior inactivation by polysulfides.

## 2. Materials and Methods

### 2.1. Recombinant Proteins

A construct encoding the catalytic active site of human PTP1B (residues 1–322 of NCBI Reference Sequence NP_002818.1) was subcloned into a pD441–H6 vector, with expression of PTP1B, purification, and tag removal performed under reducing conditions as described before [11,26,27]. Production and purification of recombinant human Trx1, TRP14, and TrxR1 wild type proteins were performed as previously described [27,28,29]. Prior to all experiments a buffer exchange ensured removal of any reductant in the enzyme preparations using Zeba Spin Desalting Columns (Thermo Scientific #87766, Waltham, MA, USA). Protein concentrations were determined using Bradford reagents (Biorad, Hercules, CA, USA).

### 2.2. Inactivation of Recombinant PTP1B Using Polysulfide or HCO_3_^−^/H_2_O_2_

Reduced recombinant PTP1B was desalted and then exposed for 30 min to either 100 μM polysulfide (Na_2_S_3_ from Dojindo Laboratories, Kumamoto, Japan, from a stock solution of 10 mM prepared freshly in degassed water and with the final concentrations in the experiments referring to those calculated for the Na_2_S_3_ salt) or 100 μM H_2_O_2_ together with 25 mM HCO_3_^−^ in assay buffer (20 mM HEPES, 100 mM NaCl buffer pH 7.4 containing 0.1 mM EDTA, 0.05% bovine serum albumin (BSA), 1 mM sodium azide). After this treatment, PTP1B was again desalted using Zeba Spin Desalting Columns to remove oxidants from the buffer, with subsequent determination of protein concentration using Bradford reagent. Different preparations of PTP1B can display different basal activities but are nonetheless here given in absolute turnover numbers and within each experiment, the treated samples should be compared to the controls in the same experiment. For protection or reactivation experiments, either pre-reduced or pre-oxidized PTP1B was treated or incubated together with the combinations of TrxR1/Trx1/TRP14 and NADPH (Sigma-Aldrich, St. Louis, MO, USA) as described in the text, with PTP1B activity finally determined in aliquots taken at 5, 15, and 60 min.

### 2.3. PTP Activity Assay

The activity of PTP1B was measured using a chromogenic substrate 4-nitrophenyl phosphate (pNPP) (P4744-1G, Sigma-Aldrich) (15 mM) as reported previously [30]. The increase in absorbance by time was measured spectrophotometrically at 410 nm and 22 °C using a Tecan Infinite M200 Pro plate reader (Tecan Group Ltd., Männedorf, Switzerland) with enzyme activity determined from the rate of absorbance increase. Substrate turnover (mol substrate converted per mol PTP1B per minute) was calculated using a 4-nitrophenol standard curve made and measured in the same volumes in the Tecan reader.

## 3. Results

### 3.1. PTP1B Inactivation by H_2_O_2_ is Potentiated by HCO_3_^−^ but that of Polysulfides is Not

We first assessed the potency of pure PTP1B inactivation by H_2_O_2_ in comparison to the inactivation by polysulfides, with or without addition of HCO_3_^−^. Confirming what we [11] and others [12] have shown earlier, we found that HCO_3_^−^ strongly potentiates the otherwise rather slow H_2_O_2_-dependent inactivation of PTP1B, but HCO_3_^−^ is not inactivating PTP1B in the absence of H_2_O_2_ as here easily demonstrated by removal of H_2_O_2_ using addition of catalase (Figure 1). Importantly, we found that inactivation of PTP1B by polysulfides was not at all affected by the presence of HCO_3_^−^, with or without addition of catalase, thus showing that polysulfides as such inhibit PTP1B by a manner not involving H_2_O_2_-related effects. The polysulfide-mediated inhibition was also potent, as reflected by the fact that 8 µM polysulfide was found to inactivate PTP1B at a similar potency as that seen using 30 μM H_2_O_2_ in the presence of HCO_3_^−^ (Figure 1).

### 3.2. Trx1 as well as TRP14 Can Potently Protect PTP1B from Inactivation by Polysulfides but Not from H_2_O_2_ in Combination with HCO_3_^−^

We next characterized the effects of Trx1 or TRP14, coupled with TrxR1 and NADPH for support of their enzymatic activities, on the effects by time on PTP1B activity during exposure to either polysulfides or to H_2_O_2_, with or without combination with HCO_3_^−^. For these experiments, we used concentrations of 250 nM TrxR1 and 10 µM Trx1 or TRP4. We also chose treatments with 80 µM H_2_O_2_ combined with 25 mM HCO_3_^−^ or 20 µM polysulfide, as those concentrations were found to inactivate PTP1B with similar potency, resulting in approximately 20–30% activity remaining upon incubation for 15 min, and less than 10% PTP1B activity remaining after 30 min of incubation. Previously we found that inhibition is due to direct effects on PTP1B and not due to inhibitory effects of the treatment on the Trx system itself [11]. Using the same experimental setup, we here found that in the presence of an active TRP14 or Trx system, the polysulfide-mediated PTP1B inactivation was instead completely abrogated. This clearly contrasted the effects seen with H_2_O_2_ combined with HCO_3_^−^ that still gave potent inactivation of PTP1B under the same conditions, with only limited protection by the Trx-system (Figure 2A,B).

There was some protection of PTP1B inactivation by polysulfides in the presence of only TrxR1 and NADPH (Figure 2C, red curve with diamonds). It was however clear that the protective effect was not only due to protection exerted by TrxR1 itself, as illustrated with the inclusion of a concentration gradient of TRP14 in protection of PTP1B treated with polysulfide (Figure 2C).

### 3.3. Reactivation of Oxidized PTP1B is More Efficient with Trx1 Than with TRP14 and More EFficient for H_2_O_2_- and HCO_3_^−^-Inactivated PTP1B Than for Polysulfide-Inactivated PTP1B

Both Trx1 and TRP14 have previously been shown to be able to reactivate PTP1B subsequent to its inactivation by reaction with either H_2_O_2_ or polysulfides [11,22,23,24]. Here we compared side-by-side the reactivation of recombinant PTP1B, by either Trx1 or TRP14, where PTP1B had first been inactivated by either H_2_O_2_ in combination with HCO_3_^−^, or by polysulfide treatment. For this, we first treated pre-reduced PTP1B with the respective oxidants, then purified the oxidatively inactivated PTP1B using desalting, and finally used that inactivated enzyme as substrate for reactivation assays. Using this setup, we found that 10 µM Trx1 coupled with 2.5 µM TrxR1 and 300 µM NADPH was able to rather efficiently reactivate PTP1B that had been inactivated by H_2_O_2_ together with HCO_3_^−^, at about half the speed of the reactivation seen using 2 mM DTT. Reactivation with 10 µM TRP14 was in this case less efficient than that seen using 10 µM Trx1 under the same conditions (Figure 3).

Assessing reactivation of polysulfide-inactivated PTP1B, we found this to be less efficient compared to the reactivation of H_2_O_2_ + HCO_3_^−^ -treated PTP1B, with 10 µM Trx1 displaying only about 20% efficiency compared to DTT and reactivation by TRP14 in this case being negligible (Figure 4).

To also test TRP14-mediated reactivation in a different approach, we exposed reduced PTP1B to 80 µM polysulfide, and then introduced an active TRP14-linked enzymatic system (10 µM TRP14 with 2 µM TrxR1 and 2 mM NADPH) two minutes prior to activity measurements at either 5, 15, or 30 min of incubation with polysulfide. This revealed that the TRP14 system could almost completely block further time-dependent polysulfide-mediated inactivation of PTP1B when added to the system, and some reactivation of activity was also found when TRP14 was added at the earlier time points (Figure 5).

## 4. Discussion

Growth factor-induced reversible oxidation and thereby inhibition of PTPs is a necessary prerequisite for downstream phosphorylation cascades to take place [31]. To enable PTP oxidation, a burst of H_2_O_2_ produced by activated NOX enzymes needs to overcome the antioxidant defense systems and reductive enzymes that protect PTPs, including the activities of TrxR1-dependent Trx, TRP14, and Peroxiredoxins [4,5,11,23]. Recent findings by us highlighted the necessity for cellular HCO_3_^−^ to be available during EGFR signaling in cells [11], likely because HCO_3_^−^ and H_2_O_2_ generates peroxymonocarbonate (HCO_4_^−^) that potently inactivates PTP1B [11,12]. Recently, we also identified that treatment with polysulfides leads to an inhibitory persulfidation of the catalytic Cys residue of PTP1B, likely serving as another mechanism for modulation of cellular growth factor responsiveness, as well as being a protective mechanism against overoxidation, with the persulfidation being both counteracted and reversed by Trx or TRP14 [22]. In the work presented here, we compared the potency of these two major mechanisms of PTP1B inactivation, i.e., through oxidation by either H_2_O_2_ + HCO_3_^−^ or by polysulfide treatment, and we evaluated how either Trx1 or TRP14 (both coupled with TrxR1 and NADPH) could maintain PTP1B in an active state when challenged with these different oxidation routes. With HCO_3_^−^ concentrations being affected by pH alterations and thereby indirectly linking PTP1B regulation to physiological acid-base balance disturbances, as discussed previously [11], we here performed all experiments at a fixed pH of 7.4 in order to facilitate these comparisons in the effects on PTP1B. We believe that some potentially important conclusions can be drawn from the results of these experiments.

We first validated that addition of HCO_3_^−^ enabled more efficient inhibition of PTP1B by H_2_O_2_, as found earlier [11,12], but, importantly, we also found that such HCO_3_^−^-potentiation was not seen with polysulfide-mediated PTP1B inactivation. Both with and without addition of HCO_3_^−^, as well as upon addition of catalase quenching any possible presence of H_2_O_2_, the inactivation of pure and pre-reduced PTP1B by polysulfides was very efficient. This clearly showed that polysulfide-mediated PTP1B inactivation is qualitatively different from that of H_2_O_2_-mediated PTP1B inactivation. It was previously shown that hydrogen sulfide (H_2_S) or polysulfide-treatment of PTP1B leads to inhibitory persulfidation of its catalytic Cys residue [21,22]. This oxidatively inhibited form of PTP1B was also reported to be potently reactivated by the Trx system, with a higher degree of efficiency compared to DTT [21]. It should be noted that we here used N_2_S_3_ as our polysulfide source. This will in solution easily autoxidize and disproportionate, resulting in a mixture of polysulfide compounds with different sulfur chain lengths that will be able to provoke persulfidation with addition of one or several sulfur atoms on reactive cysteine residues [32,33]. Since we also showed earlier that TRP14 counteracts the persulfidation of PTP1B, and the extent of EGFR phosphorylation after EGF stimulation, upon challenges of cells with polysulfides [22], we here wished to further characterize these reactions using recombinant proteins. Finding that the presence of active TRP14 as well as Trx1 can potently protect PTP1B from being inactivated by polysulfides, this agrees with the notion of these enzymes being important for protection of PTP1B activity under oxidative challenge by polysulfide treatment of cells. Based on our results shown herein, we believe that the most potent mechanism for protection of PTP1B under such conditions would be that Trx1 or TRP14 directly reduces the polysulfides, thereby indirectly protecting PTP1B from becoming inhibited by persulfidation. This also agrees with our earlier findings that Trx1 or TRP14 are indeed highly efficient in direct reduction of polysulfides [34]. In addition, TrxR1 was also found to directly reduce polysulfides on its own [34], which may explain the slight protective effect found here with regards to PTP1B activity when TrxR1 together with NADPH was present when polysulfides were added. In this case, it is likely that TrxR1 consumed some of the added polysulfides and thereby led to incomplete PTP1B inactivation, even if inclusion of increasing concentrations of TRP14 gave further increased protection of PTP1B. However, we earlier also reported that both TRP14 and Trx1 could reactivate persulfidated PTP1B, either from its directly persulfidated form or from its further oxidation state containing Cys-SSO_1–3_H species [22]. In our current study, we found this reactivation of polysulfide-treated PTP1B to be negligible with TRP14 while functional with Trx1, albeit at rather low activity compared to DTT-mediated reactivation. One difference to our previous report is that here we analyzed the reactivation of previously polysulfide-treated PTP1B that had subsequently been purified by desalting, while in our previous study the oxidation reactions followed by reactivation analyses were performed within a shorter time window and in the same vial [22]. Our current results thereby indicate that excessive polysulfide treatment of PTP1B can lead to a loss of activity in this enzyme that is difficult to regain with TRP14, but which Trx1 is still able to reactivate. We believe that PTP1B oxidation and the inhibition of PTP1B activity after treatment with Na_2_S_3_-derived polysulfides occurs primarily due to persulfidation of the active site Cys residue, upon its reaction with polysulfides (possibly having varying lengths of sulfur chains, although the persulfidated Cys with one additional sulfur atom is likely to be the dominant species). One can however also speculate that the active site Cys residue that has first been persulfidated with a single extra sulfur atom could possibly facilitate the formation of a sulfenylamide motif, in analogy to the reaction of a sulfenic acid form of the catalytic Cys as triggered upon exposure to H_2_O_2_ [17,18]. Finally, it is of course possible that other inhibitory modifications of PTP1B may also occur upon treatment with polysulfides. The identification of the exact modifications would require extensive investigations beyond the scope of this study, involving methods such as crystallography or detailed mass spectrometric analyses. We conclude, however, that Trx1 is able to reactivate at least some of these forms of PTP1B while TRP14 is less efficient.

These examples of qualitative differences in the oxidative inactivation of PTP1B, and its reactivation by either Trx1 or TRP14, should be considered when PTP1B activities are to be studied in a cellular context, where all these enzymes and oxidants will be present at different levels in a cell type- and growth context-dependent manner. The findings of our present study with the pure enzyme systems are summarized in Figure 6A.

## 5. Conclusions

Based upon our findings using recombinant enzyme systems, and considering our prior observations with regards to regulation of PTP1B by either H_2_O_2_ with HCO_3_^−^ or upon polysulfide treatment [11,22,23,34], we can propose a simple schematic model for the functional consequences of reductive and oxidative reactions regulating PTP1B in modulation of growth factor signaling. In this model, we propose that Trx1 and TRP14 mainly promotes PTP1B activity by (i) supporting Peroxiredoxins that counteract accumulation of H_2_O_2_, (ii) reducing and thus removing polysulfides as oxidative species, and, perhaps only to smaller extent, (iii) reactivating oxidatively inactivated PTP1B. These different regulatory pathways for PTP1B regulation are schematically summarized in Figure 6B. We also conclude that in a cellular context, the oxidation of PTP1B in conjunction with growth factor signaling is likely to require the combination of H_2_O_2_ and HCO_3_^−^, forming peroxymonocarbonate as the PTP1B-inhibiting species. This was suggested earlier [11] and with the findings in our present study that catalase addition diminished the inhibition of PTP1B upon a combination of H_2_O_2_ and HCO_3_^−^ but not upon a combination of polysulfides and HCO_3_^−^, the notion that peroxymonocarbonate is the inhibitory species in the former case becomes further supported. It is interesting to note that two recent reports also showed that formation of peroxymonocarbonate may promote inhibitory overoxidation of Prxs [35,36], which provided that such inhibition of Prxs would lead to higher H_2_O_2_ levels, also indirectly could lead to further PTP1B inhibition in a cellular context.

## Figures and Tables

**Figure 1 antioxidants-10-00111-f001:**
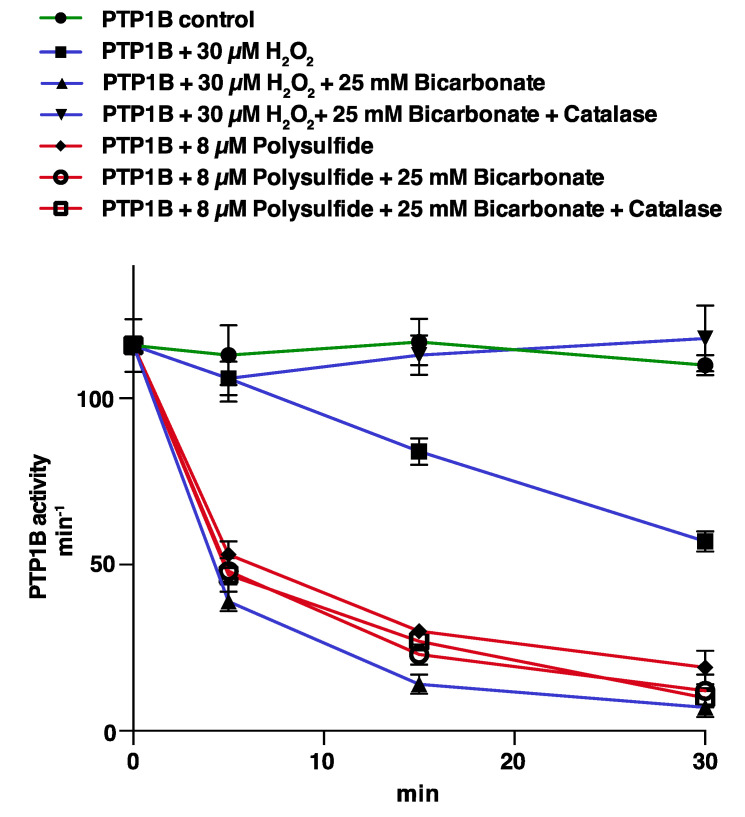
Bicarbonate/H_2_O_2_- and polysulfide-mediated inactivation of PTP1B compared. Recombinant PTP1B (600 nM) was treated with 30 µM of H_2_O_2_ (blue curves) or 8 µM polysulfide (red curves) with and without addition of 25 mM HCO_3_^−^ and catalase (20 μg/mL), as indicated in the figure. All incubations were done in 20 mM HEPES, 100 mM NaCl buffer pH 7.4 containing 0.1 mM EDTA, 0.05% bovine serum albumin (BSA) and 1 mM sodium azide, with subsequent measurements of PTP activity at the indicated times. PTP1B activity is given in min^−1^ (mol product formed/mol PTP1B/min). Data points represent means ± SEM (*n* = 4).

**Figure 2 antioxidants-10-00111-f002:**
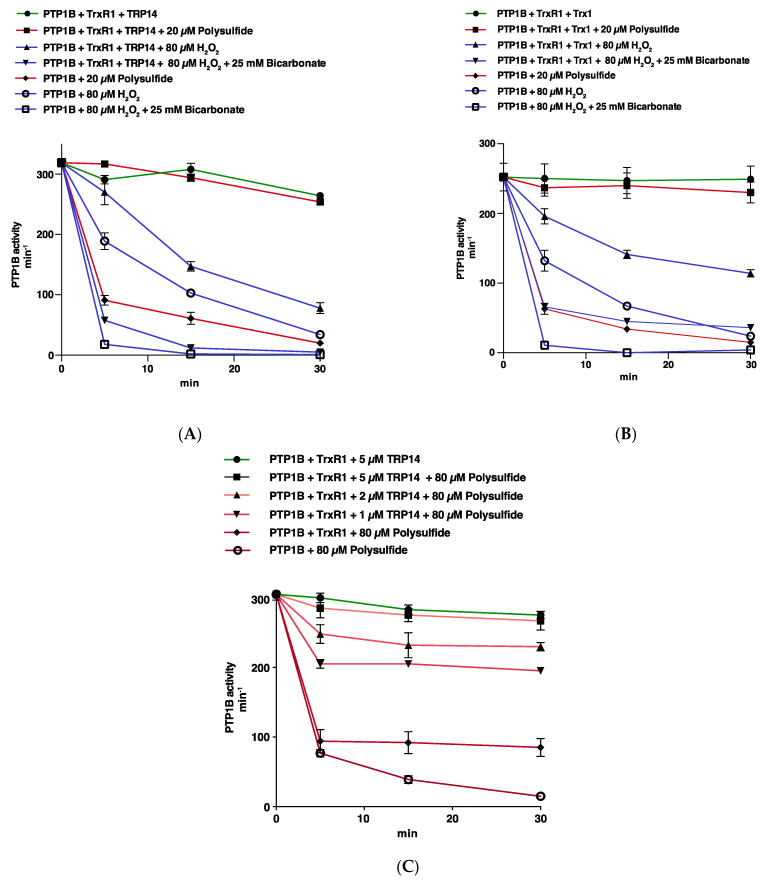
The reductive thioredoxin 1 (Trx1) or Trx-related protein of 14 kDa (TRP14) systems can prevent inactivation of PTP1B by polysulfides but not by the combination of HCO_3_^−^ and H_2_O_2_. (**A**): PTP1B (600 nM) was treated with 80 µM H_2_O_2_ (blue curves) or 20 µM polysulfide (red curves), with and without addition of 25 mM HCO_3_^−^ and the presence of TRP14 (10 µM), NADPH (300 µM), and TrxR1 (250 nM), as indicated. PTP activity was determined after 5, 15, and 30 min of incubation and data points represent means ± SEM (*n* = 3). (**B**): PTP1B was treated as in panel A but using 10 µM Trx1 instead of TRP14. (**C**): Treatment of PTP1B with polysulfide was performed as in panel A but in the presence of varying concentrations of TRP14 as indicated in the figure.

**Figure 3 antioxidants-10-00111-f003:**
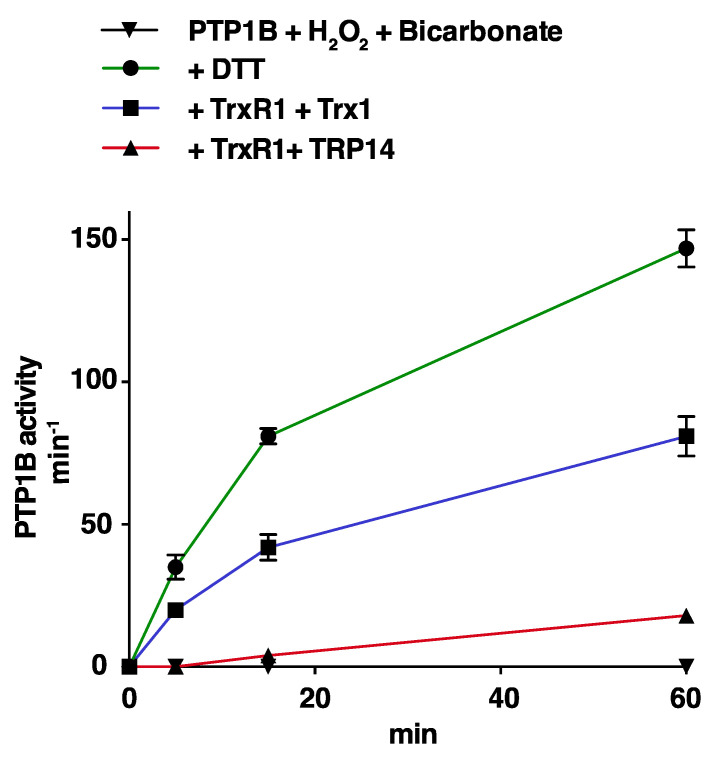
Reactivation of HCO_3_^−^/H_2_O_2_-inactivated PTP1B by the Trx1 or TRP14 systems. Recombinant PTP1B was completely inactivated using 100 µM H_2_O_2_ and 25 mM HCO_3_***^−^***. After desalting, the oxidatively inactivated PTP1B was then incubated with 2 mM DTT (green), or 10 µM of either Trx1 (blue) or TRP14 (red) together with TrxR1 (2.5 µM) and NADPH (300 µM), as indicated. PTP1B activity was then determined after incubations of 5, 15, and 60 min.

**Figure 4 antioxidants-10-00111-f004:**
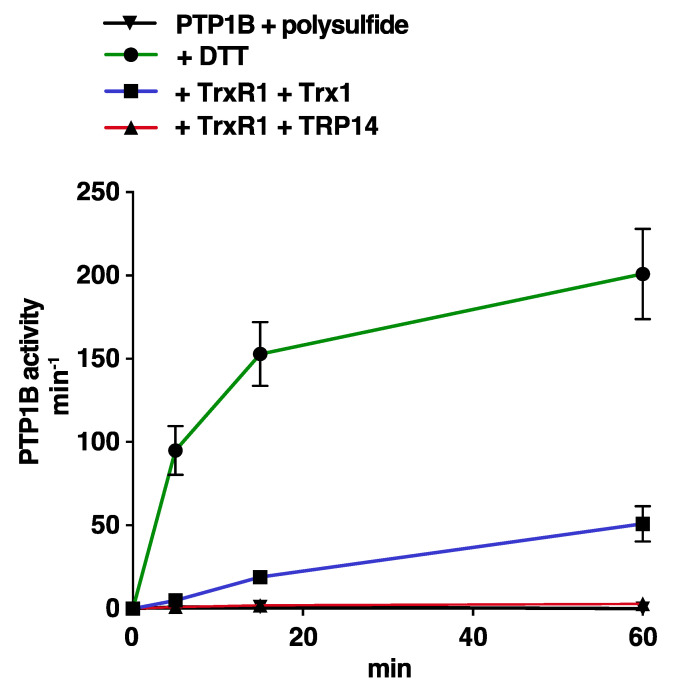
Reactivation of polysulfide-inactivated PTP1B by the Trx1 or TRP14 systems. Recombinant PTP1B was completely inactivated using incubation with 100 µM polysulfide. After desalting, the oxidatively inactivated PTP1B was then incubated with 2 mM DTT (green) or 10 µM of either Trx1 (blue) or TRP14 (red) together with TrxR1 (2.5 µM) and NADPH (300 µM), as indicated. Reactivated PTP1B activity was then determined after incubations of 5, 15, and 60 min.

**Figure 5 antioxidants-10-00111-f005:**
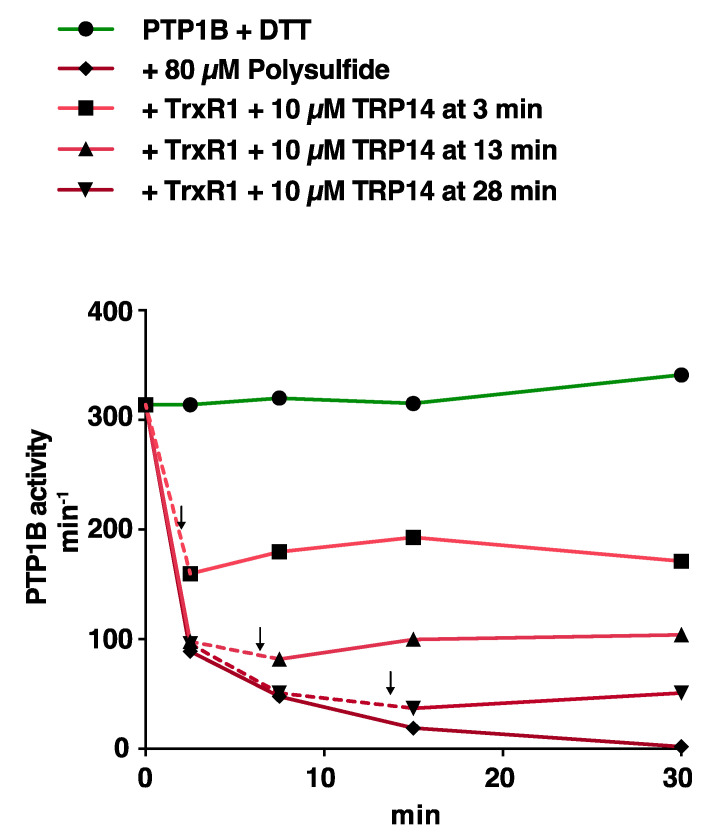
Abrogation of polysulfide-dependent inactivation of PTP1B by addition of the TRP14 redox system. PTP1B (600 nM) was incubated together with 80 µM polysulfide and aliquots were taken for activity measurements at 5, 15, and 30 min. Two minutes before each time point, addition of TRP14 (10 µM), NADPH (2 mM), and the TrxR1 (2 µM) was also performed with specific samples, as indicated with arrows and dashed parts of the curves. Data points represent means ± SEM (*n* = 3).

**Figure 6 antioxidants-10-00111-f006:**
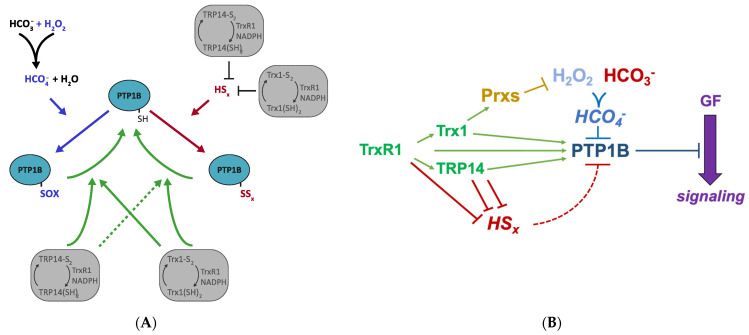
Schemes for the regulation of PTP1B activity by the Trx1 or TRP14 systems in relation to oxidation by either polysulfides or peroxymonocarbonate. (**A**) Model scheme summarizing the data and interpretations of this study, showing how PTP1B activity can be regulated by different redox dependent mechanisms through H_2_O_2_/HCO_3_^−^ and the Trx1 or TRP14 systems. The reaction of H_2_O_2_ together with HCO_3_^−^ yields peroxymonocarbonate (HCO_4_^−^), which oxidizes PTP1B to forms (“PTP1B_SOX_”), including a sulfenylamide [12], that are amenable to reactivation by either Trx1 or TRP14 (green arrows). Polysulfides (“HS_x_”, red) are potently reduced by either Trx1 or TRP14, whereby these enzyme systems indirectly prevent polysulfides from inhibiting PTP1B. Once persulfidated PTP1B has been formed (“PTP1B_SSx_”), this can mainly be reactivated by Trx1 but not as efficiently with TRP14 (dashed green arrow). The figure in (**B**) illustrates schematically how the activities of the Trx/TRP14/TrxR1 system together with Peroxiredoxins (Prxs) can maintain PTP1B activity in a cellular context, with Trx1-depdendent Prxs counteracting H_2_O_2_ accumulation, both Trx1 and TRP14 reducing polysulfides, and both systems being able to reduce and thereby reactivate oxidatively inactivated PTP1B. From our results in the present study, we conclude that polysulfide-mediated inhibition of PTP1B (dashed red arrow) must overcome the very potent direct reduction of polysulfides seen with the Trx1 and TRP14 systems, while H_2_O_2_ together with HCO_3_^−^ yielding peroxymonocarbonate (HCO_4_^−^) can inhibit PTP1B also in the presence of active Trx1 and TRP14 systems. Ultimately the extent of PTP1B activity will control growth-factor (GF) signaling, with more inhibited PTP1B allowing for more signaling activity.

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
