# Peer review of "Qualitative Differences in Protection of PTP1B Activity by the Reductive Trx1 or TRP14 Enzyme Systems upon Oxidative Challenges with Polysulfides or H2O2 Together with Bicarbonate"

_antioxidants, 2021, doi:10.3390/antiox10010111_

Round 1

Reviewer 1 Report

In the present manuscript, Dagnell, Cheng and Arnér analyzed the inhibitory effects of polysulfides on the catalytic activity human Protein tyrosine phosphatase 1B (PTP1B) and compared them with the inhibition of the enzyme by H2O2 and HCO3-. They also compared the protective effects xerted by thioredoxin-1 (Trx1) and a thioredoxin-related protein of 14 kDa (TRP14) against the inhibition of PTP1B by polysulfides and H2O2/HCO3-.

It turns out that

  • polysulfides are more effective in inactivating pure PTP1B than the combination H2O2/HCO3-;
  • HCO3- alone has no effect on PTP1B, also in combination with polysulfides, confirming that that bicarbonate-assisted H2O2-mediated inactivation of PTP1B involves formation of peroxymonocarbonate;
  • both Trx1 and TRP14 efficiently protect PTP1B from inactivation by polysulfides, but less so by H2O2/HCO3-;
  • Trx1 is more efficient than TRP14 in reactivating oxidized PTP1B;
  • polysulfide-mediated PTP1B inactivation sensibly differs from that mediated by H2O2/HCO3-;
  • Trx1 or TRP14 directly reduce the polysulfides in vivo, thereby indirectly protecting PTP1B.

This is a very well written and interesting paper, addressing a very attractive topic. The experiments were well performed and the obtained data were clearly presented and thoroughly discussed, providing relevant findings. Hence, the scope and content of the manuscript perfectly match those of Antioxidant. In my opinion the present version of the manuscript by Dagnell, Cheng and Arnér requires minor revisions to be accepted for publication in Antioxidant.

  1. The authors should explain how the catalytic site of PTP1B changes upon reaction with
  2. Can the authors provide any hint about the molecular details of mechanism of the reaction between PTP1B and peroxymonocarbonate?
  3. Does pH influence the latter reaction?

Author Response

Response:

We thank the reviewer for finding our study to be well written and interesting, and for the excellent summary of our main findings. We also thank the reviewer for the good suggestions for further improvements, outlined as follows.

Comment:

  1. The authors should explain how the catalytic site of PTP1B changes upon reaction with

Response:

We note that this comment is missing a few words at the end of the sentence. Since question nr 2 addresses the mechanism of peroxymonocarbonate-derived inhibition of PTP1B we assume that this first comment asked us to further evaluate the mechanisms of how polysulfides may react with PTP1B.

The polysulfide compound that we used was the commercially available polysulfide salt Na2S3. When dissolved in buffer, autooxidation and disproportionation events easily yield mixtures of polysulfide compound derivatives containing sulfur chains of different lengths, which may react with accessible thiolates of proteins to produce persulfidated Cys residues (PMID 27697462; PMID 25747475). When we state a concentration of “polysulfide” in the paper, it refers to that calculated from the prepared stock solution of the Na2S3 salt, but it cannot be known what exact proportion of the resultant different sulfane sulfur compounds that are actually present in the experiments. We believe that PTP1B oxidation and inhibition of PTP1B activity after treatment with a Na2S3 stock solution occurs primarily due to persulfidation of the active site Cys-215, upon its reaction with polysulfide compounds (possibly having varying lengths of sulfur chains, although the persulfidated Cys with one additional sulfur atom is likely to be the dominant species of these reactions). One can however also speculate that the active site Cys residue that first becomes persulfidated with a single extra sulfur atom could, possibly, facilitate the formation of a sulfenylamide motif, in analogy to that reaction easily occuring with the reactivity of a sulfenic acid form of the catalytic Cys produced upon exposure to H2O2. Finally, it is of course possible that other inhibitory modifications of PTP1B may also occur but the identification of the exact modifications would require extensive investigations beyond the scope of this study, involving methods such as crystallography or detailed mass spectrometric analyses. To clarify and better address these questions, we have now made the following additions to the text:

Methods, lines 96-98, better specifying how the polysulfide salt was prepared and used by adding the following words:

“…, , from a stock solution of 10 mM prepared freshly in degassed water and with the final concentrations in the experiments referring to those calculated for the Na2S3 salt)”.

Discussion, lines 251-254:

“It should be noted that we here used N2S3 as our polysulfide source. This will in solution easily autoxidize and disproportionate, resulting in a mixture of polysulfide compounds with different sulfur chain lengths that will be able to provoke persulfidation with addition of one or several sulfur atoms on reactive cysteine residues [35,36].”

Discussion, lines 279-290:

We believe that PTP1B oxidation and the inhibition of PTP1B activity after treatment with Na2S3-derived polysulfides occurs primarily due to persulfidation of the active site Cys residue, upon its reaction with polysulfides (possibly having varying lengths of sulfur chains, although the persulfidated Cys with one additional sulfur atom is likely to be the dominant species). One can however also speculate that the active site Cys residue that has first been persulfidated with a single extra sulfur atom could possibly facilitate the formation of a sulfenylamide motif, in analogy to the reaction of a sulfenic acid form of the catalytic Cys as triggered upon exposure to H2O2 [18]. Finally, it is of course possible that other inhibitory modifications of PTP1B may also occur upon treatment with polysulfides. The identification of the exact modifications would require extensive investigations beyond the scope of this study, involving methods such as crystallography or detailed mass spectrometric analyses. We conclude, however, that Trx1 is able to reactivate at least some of these forms of PTP1B while TRP14 is less efficient.” 

  1. Can the authors provide any hint about the molecular details of mechanism of the reaction between PTP1B and peroxymonocarbonate?

Response:

In a previous study by Gates et al they investigated the effects of the combination of sodium bicarbonate and H2O2, and the resulting product peroxymonocarbonate, with regards to the exact effects on PTP1B. In their study the oxidation product that was found using crystallographic analyses revealed formation of a sulfenylamide of the active site cysteine. Control experiments using chelators suggested that the oxidation occured through a two electron transfer mechanism (Zhou et al J Am Chem Soc, 2011,PMID 21913686). We have now included this clarification in the figure legend of Fig. 6, line 313:

“, including a sulfenylamide [12],

  1. Does pH influence the latter reaction?

Response:

This is a good and relevant question also because bicarbonate is a major physiological buffer, and its concentration in cells will thus change upon acid-base fluctuations. This possibility was addressed in our previous study on the effects on the combinations of bicarbonate and H2O2 on PTP1B, where we also showed how different lactic acid concentrations modulated the PTP1B-modulated cellular growth-factor signaling, and we thoroughly discussed the importance of pH regulation with regards to these effects (Dagnell et al, reference 11 in the present manuscript). In the experiments performed in the current study, we aimed at directly comparing the effects of peroxymonocarbonate on PTP1B activity with those of polysulfides under the same conditions. Therefore, sodium bicarbonate was here dissolved in pH-buffered HEPES and the pH was set to 7.4 in all experiments prior to performing the activity analyses, as clearly described in the methods section (line 99). To further emphasize this fact in the paper, we have now included the following sentence in the Discussion on lines 237-240:

“With HCO3- concentrations being affected by pH alterations and thereby indirectly linking PTP1B regulation to physiological acid-base balance disturbances, as discussed previously [11], we here performed all experiments at a fixed pH of 7.4 in order to facilitate these comparisons in the effects on PTP1B. “

Reviewer 2 Report

The manuscript by Dagnell et al. faces the mechanism for reversible oxidation and consequent inactivation of the recombinant protein tyrosine phosphatase 1B (PTP1B), a key component involved in the activation of downstream phosphorylation cascades during cell growth. Two different oxidation mechanisms were investigated, namely that caused by polysulfides or by a co-treatment with hydrogen peroxide and bicarbonate; also the protective role exerted by two reducing enzyme systems, constituted by Trx1 or Trx-related protein TRP14, both coupled with TrxR1 and NADPH was considered. The study revealed that hydrogen peroxide inactivation of PTP1B is strongly potentiated by bicarbonate and probably occurs through a mechanism different from that observed with polysulfides; both thioredoxin systems protect PTP1B from inactivation by polysufides, whereas they are much less effective in the hydrogen peroxide plus bicarbonate inactivation. Furthermore, the Authors investigated the possible reactivation of an inactivated PTP1B form and found a greater efficiency with the enzyme inactivated by hydrogen peroxide plus bicarbonate, the reducing system containing TRP14 being thoroughly less efficient. However, when added during a polysulfide inactivation experiment, the TRP14-based reducing system is capable to block a further oxidation of PTP1B.

The manuscript is well organised and the usage of a purified recombinant form of PTP1B is relevant for the appropriate biochemical analysis of the effects of oxidant/reducing systems. Therefore, the article could merit publication in Antioxidants provided that the following minor changes are introduced in it.

Minor points

-In the formula of bicarbonate the negative charge seems to be misplaced; please try to edit it as an apex of the formula, not of “3”; a similar editing should be applied for peroxymonocarbonate.

-Fig. 1. Explain the reasons for replacing an obvious experiment with bicarbonate alone by that containing 30 µM H2O2+ 25 mM bicarbonate + catalase.

-The Authors suggest that the mechanism for the oxidation of PTP1B by H2O2and bicarbonate should involve the formation of the stronger oxidant peroxymonocarbonate. Please, discuss if the target residue for this oxidation is the same Cys oxidised by polysulfides. Furthermore, when comparing rates of PTP1B inactivation in Fig. 1 in different experimental conditions, a low rate is observed also with H2O2alone. Because of the absence of bicarbonate, what is the possible mechanism for this oxidation?

-Results, page 4, lines 144-146. The statement “… which clearly contrasted that seen with H2O2combined with HCO3 that gave potent inactivation of PTP1B under the same conditions” could be misinterpreted. Please, comment the data in a more appropriate way, by stating that the two reducing systems display a significantly lower, although detectable, protection from inactivation by H2O2+ HCO3, mainly with the Trx1-based system.

-Fig. 2C. Check for color coding of lines and symbols in this panel.

-Results, page 6, lines 157-159. The statement “… the protective effect was concentration dependent with regards to TrxR1 redoxin substrate, … … using a concentration gradient of TRP14 with PTP1B treated with polysulfide” could be misleading. Please, use a more direct sentence, stating that the protective effect was dependent on TRP14 concentration.

Author Response

Response:

We thank the reviewer for finding our paper to be well organized and to merit publication in Antioxidants, and also thank the reviewer for the good suggestions for further improvements, outlined as follows.

Minor points

-In the formula of bicarbonate the negative charge seems to be misplaced; please try to edit it as an apex of the formula, not of “3”; a similar editing should be applied for peroxymonocarbonate.

Response:

Thank you, we agree that the negative charge symbol superscript looks as if it is misplaced, but actually it is not. It seems as if the font of the Antioxidant Word template file leads to this appearance. We will make sure during proof reading of the final type-set manuscript that sub- and superscript symbols appear as they should.

-Fig. 1. Explain the reasons for replacing an obvious experiment with bicarbonate alone by that containing 30 µM H2O2+ 25 mM bicarbonate + catalase.

Response:

This is indeed an obvious control experiment, but it was not included here because that and many more control experiments were done in our previous paper reporting our first analyses of the effects of H2O2+bicarbonate on PTP1B (reference 11 in the manuscript). We refer to that study on lines 118-121, but now we note that the easily made catalase inclusion was specifically utilized in this paper by introducing the word “here” on line 120.

-The Authors suggest that the mechanism for the oxidation of PTP1B by H2O2and bicarbonate should involve the formation of the stronger oxidant peroxymonocarbonate. Please, discuss if the target residue for this oxidation is the same Cys oxidised by polysulfides. Furthermore, when comparing rates of PTP1B inactivation in Fig. 1 in different experimental conditions, a low rate is observed also with H2O2alone. Because of the absence of bicarbonate, what is the possible mechanism for this oxidation?

Response:

Regarding the mechanisms of inactivation by the different treatments, please see above our responses to points 1 and 2 raised by reviewer 1. With regards to the (rather low) rate of inactivation of PTP1B by H2O2 alone, this is likely due to induction of a sulfenic acid derivative of the catalytic Cys residue that subsequently converts to a sulfenylamide, as discussed at further lengths in references 11, 12, 18 and several of the cited reviews. To emphasize this fact, we have now included the words “otherwise rather slow” on line 119.

-Results, page 4, lines 144-146. The statement “… which clearly contrasted that seen with H2O2combined with HCO3– that gave potent inactivation of PTP1B under the same conditions” could be misinterpreted. Please, comment the data in a more appropriate way, by stating that the two reducing systems display a significantly lower, although detectable, protection from inactivation by H2O2+ HCO3–, mainly with the Trx1-based system.

Response:

Thank you, these statements have been revised for improved clarity and now read as follows (lines 146-150):

“Using the same experimental setup we here found that in the presence of an active TRP14- or Trx-system, the polysulfide-mediated PTP1B inactivation was instead completely abrogated. This clearly contrasted the effects seen with H2O2 combined with HCO3- that still gave potent inactivation of PTP1B under the same conditions, with only limited protection by the Trx-system (Fig. 2A and 2B).”

-Fig. 2C. Check for color coding of lines and symbols in this panel.

Response:

We have checked the color coding of Fig. 2C, and believe it is correct. We have consistently throughout the paper used green for non-treated controls, blue for H2O2 combined with HCO3- and red for polysulfide treatments. Since Fig. 2C shows only polysulfide treatment all curves are red, but in different shades in order to reflect the varying concentrations of TRP14.

-Results, page 6, lines 157-159. The statement “… the protective effect was concentration dependent with regards to TrxR1 redoxin substrate, … … using a concentration gradient of TRP14 with PTP1B treated with polysulfide” could be misleading. Please, use a more direct sentence, stating that the protective effect was dependent on TRP14 concentration.

Response:

Tank you, this has now been revised for improved clarity and the sentence now reads as follows (lines 169-174):

“There was some protection of PTP1B inactivation by polysulfides in the presence of only TrxR1 and NADPH (Fig. 2C, red curve with diamonds). It was however clear that the protective effect was not only due to protection exerted by TrxR1 itself, as illustrated with the inclusion of a concentration gradient of TRP14 in protection of PTP1B treated with polysulfide (Fig. 2C).”

Reviewer 3 Report

In the paper: “Qualitative Differences in Protection of PTP1B Activity by the Reductive Trx1 or TRP14 Enzyme Systems upon Oxidative Challenges with Polysulfides or H2O2 together with Bicarbonate” Dagnell M. and co-Authors compared the potency of inactivation by bicarbonate / H2O2 with that of Na2S2.

The principal findings reported are the more efficient inhibition of the purified protein by Na2S2 as compared to the combination of bicarbonate and H2O2 counteracted by either TRP14 or Trx1, and the more potent PTP1B reactivation capacity of Trx1 with respect to TRP14.

The article is well written and the aims of the various experiments are clearly explicated.

However, I think that major issue still need to be addressed.

First, throughout the paper Authors claim that the effect of polysulfides on PTP1B is studied, but the only polysulfide used is sodium disulfide (Na2S2). Reference is made to other papers where other polysulfides were utilized. In fact, compounds like Na2S3 or Na2S4 are commercially available. Thus, I would specify the species employed in the experiments without being so general otherwise please present also the data obtained with other polysulfides.

Next, Authors report that the Trx1 and TRP14 systems are not able to prevent PTP1B inhibition by bicarbonate / H2O2 (Figure 2 a and b). I would suggest to check whether Trx and TRP14 themselves are sensitive to H2O2 t the concentrations used in their experiments. Maybe the oxidation of cysteines of Trx1 and TRP14 active sites is involved in the inefficiency of the prevention of PTP1B inhibition.

Finally, in Figure 2c, TrxR1 alone can protect PTP1B from further inhibition at 15 and 30 min. How can it do that in the absence of Trx1 or TRP14? Please add a comment on that. For the same reason, in Figure 5 please add the effect on PTP1B activity by TrxR alone. Indeed, there is the possibility that the addition of additional proteins to the reaction mixture prevents further oxidation because of a competitive effect with PTP1B for the activity of the oxidant.

In conclusion, I think that the paper has some critical points and needs major revisions. I invite the Authors to perform the various controls that will let them have a more complete picture of the mechanisms under investigation.

Author Response

Response:

We would like to thank the reviewer for finding our paper well written, and for valuable suggestions to make the manuscript clearer to the reader.

First, throughout the paper Authors claim that the effect of polysulfides on PTP1B is studied, but the only polysulfide used is sodium disulfide (Na2S2). Reference is made to other papers where other polysulfides were utilized. In fact, compounds like Na2Sor Na2S4 are commercially available. Thus, I would specify the species employed in the experiments without being so general otherwise please present also the data obtained with other polysulfides.

Thank you for noting this important point. First, unfortunately there was a typo in the previous manuscript as we indeed used Na2S3 and not Na2S2. This typo has now been corrected. Second, there are indeed different variants of commercially available polysulfide compounds. As discussed in our response to point nr 1 of reviewer 1, others have previously shown and discussed that polysulfide compounds easily autoxidize or disproportionate, yielding compounds with several different sulfur chain lengths in solution, making it hard if not impossible to know exactly what compounds that are the active species in the experiment, although most or all of such polysulfide derivatives are likely to yield persulfidation products on susceptible Cys residues. This is now better discussed in the revised version of the paper (see above, response to point 1 of reviewer 1).

Next, Authors report that the Trx1 and TRP14 systems are not able to prevent PTP1B inhibition by bicarbonate / H2O2 (Figure 2 a and b). I would suggest to check whether Trx and TRP14 themselves are sensitive to H2O2 t the concentrations used in their experiments. Maybe the oxidation of cysteines of Trx1 and TRP14 active sites is involved in the inefficiency of the prevention of PTP1B inhibition.

Response:

As highlighted and noted by the reviewer this is of course a very important control. In our previous study (reference 11 in the manuscript) we thereby naturally tested if the Trx system would be affected in any manner by this treatment, using the Trx-dependent insulin reduction assay measuring NADPH consumption. In such controls, we have not detected any effects (see Suppl. Fig. S1 in ref. 11). We have now clarified this with the following statement on lines 145-146:

“Previously we found that this inhibition is due to direct effects on PTP1B and not due to inhibitory effects of the treatment on the Trx system itself [11].”

Finally, in Figure 2c, TrxR1 alone can protect PTP1B from further inhibition at 15 and 30 min. How can it do that in the absence of Trx1 or TRP14? Please add a comment on that. For the same reason, in Figure 5 please add the effect on PTP1B activity by TrxR alone. Indeed, there is the possibility that the addition of additional proteins to the reaction mixture prevents further oxidation because of a competitive effect with PTP1B for the activity of the oxidant.

Response:

We showed previously that polysulfides are direct substrates for TrxR1 (ref. 37). We thus certainly agree with the reviewer in believing that protection could be through competitive effects due to enzymatic clearance of the added polysulfides. We have now clarified this point by adding the following two text sections:

Results, lines 169-170:

“There was some protection of PTP1B inactivation by polysulfides in the presence of only TrxR1 and NADPH (Fig. 2C, red curve with diamonds).”

Discussion, lines 264-269:

“In addition, TrxR1 was also found to directly reduce polysulfides on its own [37], which may explain the slight protective effect found here with regards to PTP1B activity when TrxR1 together with NADPH was present when polysulfides were added. In this case, it is likely that TrxR1 consumed some of the added polysulfides and thereby led to incomplete PTP1B activation, even if inclusion of increasing concentrations of TRP14 gave further increased protection of PTP1B.”

In conclusion, I think that the paper has some critical points and needs major revisions. I invite the Authors to perform the various controls that will let them have a more complete picture of the mechanisms under investigation.

Response:

Thank you for the good comments. We hope that our responses to this reviewer, and to the two other reviewers, with the corresponding revisions of the text have led to a more complete picture of the mechanisms under investigation. We certainly think the paper has been improved as a result of this revision and we would wish to thank all three reviewers for their constructive comments.

Round 2

Reviewer 3 Report

The paper has been modified in accordance with the points raised by the first revision process and improved especially in the discussion.

I do not have further questions, therefore, I think that the paper is now suitable for publication in Antioxidants.